# Combination Optimization Configuration Method of Capacitance and Resistance Devices for Suppressing DC Bias in Transformers

**Donghui Wang and Chunming Liu \***

School of Electrical and Electronic Engineering, North China Electric Power University, Beijing 102206, China; personalintouch@foxmail.com

**\*** Correspondence: liuchunming@ncepu.edu.cn

**Abstract:** The ground current of a high-voltage direct current (HVDC) transmission system can cause DC bias in transformers near the grounding electrode during monopole operations, which affects the alternating current (AC) power system operation. Owing to multiple bias current flow paths, a capacitance blocking device installed at the neutral point of a transformer may increase the DC bias in adjacent transformers, while suppressing the DC current in that transformer. This paper introduces the use of an effective bias current indicator to describe the effect of the grounding current on transformers in a network, considering the wiring characteristics of the autotransformers and the power system topology. Additionally, a combination optimization method for the capacitance and resistance is applied in order to determine the minimum number of installed devices that restrict the maximum effective bias current throughout the network to a permissible range. A genetic algorithm based on an improved roulette selection method is adopted to solve the optimal configuration problem. The method is validated by using a test case based on the Xizhe HVDC transmission receiving-end grid near the Jinsi grounding electrode. The configuration of the capacitance and resistance was optimized by the improved genetic algorithm. This method can achieve the desired level of DC bias management with fewer devices than the conventional method, which verifies the feasibility and superiority of the proposed optimization method.

**Keywords:** capacitance blocking device; DC bias; optimal configuration; resistance current limiting device

## 1. Introduction

When a high-voltage direct current (HVDC) transmission system operates in monopole mode, a large DC current flows into the Earth through the grounding electrode, which results in a DC bias in the transformers near the grounding electrode [1–3]. The configuration of such an HVDC transmission systems was studied by the authors of [4]. DC bias has a series of adverse effects on the safe operation of transformers and AC power grids [5–8], such as increased transformer temperatures, vibration, and noise, and a greater generation of harmonics and grid reactive power fluctuations [9,10]. This can endanger the safe and stable operation of the reactive power compensation equipment by causing control and protection device malfunction and refusal [11,12].

The methods for suppressing DC bias in transformers mainly include mounting a capacitance blocking device (CBD) mounting or resistance current limiting device (RCLD) at the neutral point, current injection, and neutral-point potential compensation. A method for optimizing CBD installation to control the DC bias of a transformer was analysed in the literature [13,14]. The influence of adding a resistance below 10 Ω to the neutral point of the main transformer in stations was discussed by the authors of [15,16], and Dutta and Bhattacharya researched a current injection method through

simulation calculations and online monitoring [17]. A potential compensation method for restraining the DC bias of autotransformers is discussed in the literature [18,19]. These previous works generally use the neutral-point current to describe the impact of the grounding current on the transformer. In addition, these studies focused on individual transformer DC bias, ignoring system-level analysis. Nevertheless, current systems broadly adopt autotransformers, in which the DC current remains in the series winding after the neutral-point processed. Therefore, it is not reasonable to use the neutral-point current as an indicator of DC bias. Moreover, at the system level, the conventional method is generally used to install management devices at the transformers with the largest bias current in the target network. This conventional method is likely to increase the bias current in the adjacent transformers beyond their limits, thus intensifying the DC bias of the overall network. Therefore, a reasonable indicator is essential to describing the impact of the grounding current on the transformers. Moreover, an optimization method to configure the management devices for suppressing the overall DC bias of the transformers in the network should be considered.

Based on the above premises, this paper proposes a combination optimization method to configure RCLDs and CBDs so as to control the DC bias of the transformers in the overall network. An indicator of the effective bias current of the transformers is applied to describe the influence of the DC ground current on the transformers, and the constraint condition is a limit on the effective bias currents of the transformers in the overall network. The objective is to minimize the installations of CBDs and RCLDs by applying a genetic algorithm based on an improved roulette selection method, so as to optimize the CBD and RCLD configurations. The Xizhe HVDC transmission project receiving-end grid near the Jinsi grounding electrode is taken as a test network to verify the effectiveness of the optimization method.

The main contributions and novelty of this paper are summarized as follows:

(1) We proposed the transformer effective bias current as an indicator based on the transformer structure and characteristics, in order to describe the influence of the grounding current on the transformers, which is applicable to all types of transformers. In contrast, the previous indicator, that is, the neutral-point DC current, cannot reflect the DC current in the series windings of the autotransformers.

(2) This paper proposed an optimal method that can ensure that the effective bias currents of the transformers in the overall network do not exceed certain limits. In addition, it can determine the minimum number of management devices, whereas the traditional method for suppressing the DC bias of transformers in the overall network is simply installing management devices in substations with a large bias current. We found that this conventional method would increase the bias current of the adjacent transformers.

(3) We simulated managing the transformer DC bias in the overall network with resistance devices, capacitance devices, and combination devices. Additionally, the combination optimization method based on the transformer effective bias current indicator that we established can achieve the same management level with the fewest devices. Notably, resistance devices cannot always restrict the transformer effective bias currents in the overall network to a permissible range.

The rest of the paper is organized as follows. Section 2.1 proposes a method for calculating the Earth potential distribution around a DC grounding electrode, and Section 2.2 presents the effective bias current definition and computing method. Section 2.3 discusses a method for optimizing the CBD and RCLD configuration. The designed models are verified in Section 3 using the Xizhe HVDC transmission project receiving-end grid near the Jinsi grounding electrode, and Section 4 summarizes and concludes this paper.

## 2. Methodology

### 2.1. Earth Potential Distribution Calculation Method

The ground current will flow into the Earth when an HVDC transmission system operates in monopole mode, which forms a closed loop through the Earth. This ground current will impose a

potential on the substation grounding system near the electrode because of the different soil resistivities. The calculation method of the Earth potential near the HVDC grounding electrode has been thoroughly discussed in previous works (e.g., [2,12]). The potential distribution satisfies the following Laplace equation according to the inferences of Maxwell's Equations:

$$\bigtriangledown^2 V = 0, \tag{1}$$

where $V$ represents the potential. The DC grounding electrode is generally regarded as a point current source when considering a large-scale geoelectric potential distribution. At the interface of adjacent soil $i$ and $j$ with different conductivities, the boundary conditions are given by the following:

$$V_i = V_j, \tag{2}$$

$$\frac{1}{\rho_i} \frac{dV_i}{dn} = \frac{1}{\rho_j} \frac{dV_j}{dn}, \tag{3}$$

where $V_i$ and $V_j$ are the soil potentials, $\rho_i$ and $\rho_j$ are the soil conductivities, and $n$ is the outer normal direction pointing toward the air side. The grounding electrode is used as a boundary for solving the field, considering it is a good conductor and its surface is an equipotential. After determining the field equations and boundary conditions, the ground potential distribution around the DC grounding electrode can be solved by numerical calculations using advanced commercial numerical calculation software.

## 2.2. Bias Current Calculation Method

The grounding current forms a potential in the substation grounding system near the grounding electrode, where the Earth acts as part of the DC path. The DC current flows into the alternating current (AC) system near the grounding electrode through the transformer neutral point, which causes a DC bias in the transformer [20]. Therefore, the problem becomes a circuit calculation. Previous works generally used the neutral-point DC current to characterize the severity of the transformer DC bias [21]. However, the neutral-point DC current cannot reflect the DC current in the series windings of the autotransformers (Figure 1). The autotransformer equivalent circuit is generally modelled as in Figure 1. This paper proposes the use of the transformer effective bias current to describe the influence of the grounding current on the transformers, and to determine the optimal CBD and RCLD configuration. The effective bias current $I_e$ on autotransformer $T$ (Figure 1) is defined considering its magnetic flux, as follows:

$$I_e = I_h + I_m / k_T, \tag{4}$$

$I_e$ can be calculated according to the autotransformer equivalent circuit (Figure 1), as follows:

$$I_e = \boldsymbol{\Phi_a} \boldsymbol{V_a} \tag{5}$$

$$\boldsymbol{\Phi_a} = [y_{hm} - \frac{y_{hm}}{k_T}, \frac{1}{k_T}(y_{ms} + y_{hm}) - y_{hm}, -\frac{y_{ms}}{k_T}], \tag{6}$$

$$\boldsymbol{V_a} = [V_h, V_m, V_s]^T, \tag{7}$$

where $I_h$ and $I_m$ are the DC currents flowing into the autotransformers from the HV and medium voltage (MV) sides, respectively; $k_T$ is the ratio of the HV to MV sides of the autotransformer; $V_h$, $V_m$, and $V_s$ are the potentials of HV node $h$, MV node $m$, and neutral-point node $s$, respectively, with the reference point being $s$; and $y_{hm}$ and $y_{ms}$ are the admittance of the series winding and common winding, respectively. Equations (5)–(7) are obtained from the autotransformer equivalent circuit. Equation (4) means that the transformer effective bias current $I_e$ is the sum of the DC current at both the HV side $I_h$ and at the MV side $I_m$, which should be multiplied by the ratio of turns to convert to the HV side.

According to the theory of the node admittance matrix, the current can be converted into the product of voltage and conductance. $V_a$ is the node voltage matrix, and the matrix $\Phi_a$ is the corresponding conductivity of each winding after being converted to the HV side. The derivation process is based on the circuit shown in Figure 1.

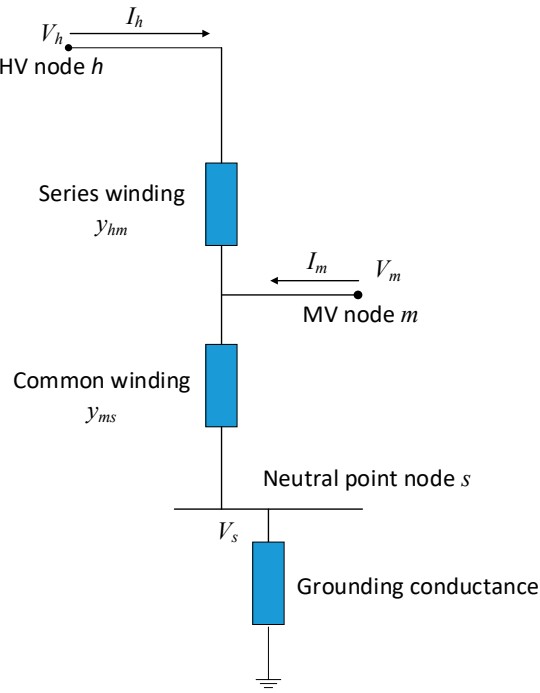

**Figure 1.** Autotransformer equivalent circuit model. HV—high voltage; MV—medium voltage.

For an ordinary transformer that is not an autotransformer, there is no electrical link between the HV side and the MV side; thus, the effective bias current is the neutral-point current. The effective bias current, $I_E$, in an ordinary transformer, $t$, (Figure 2) is defined as follows:

$$I_E = [y_{ko}, -y_{ko}][V_k, V_o]^T,　　　　　　　　(8)$$

where $V_k$ and $V_o$ are the potentials of HV node $k$ and neutral point node $o$, respectively, and $y_{ko}$ is the admittance of the winding.

When a DC current is injected into the ground through an electrode, a potential distribution forms around the electrode. The bias currents are through toward the transformers (through transmission lines) when their neutral-point potentials are different. Only the resistances need to be considered for a DC circuit. When the CBD is not installed at the neutral point, $d$, of one transformer, the current, $I_d$, injected into $d$ is as follows:

$$I_d = \frac{V_d}{R_d},　　　　　　　　(9)$$

where $V_d$ represents the ground potential at node $d$ caused by the grounding electrode, and $R_d$ is the ground resistance corresponding to node $d$. The relationship between the node voltage vector, $V$, and the node injection current vector, $I$, is as follows:

$$V = Y^{-1}I　　　　　　　　(10)$$

The common winding current will be 0 if a capacitance device is installed at the neutral point of autotransformer $T$, but the series winding current will not be 0. Then, $I_e$ is given by the following:

$$I_e = (V_h - V_m)y_{hm},　　　　　　　　(11)$$

where $y_{hm}$ is the series winding conductance. If the neutral point of autotransformer $T$ is not equipped with a capacitance device, $I_e$ is given by the following:

$$I_e = (V_h - V_m)y_{hm} + \frac{(V_m - V_s)y_{ms} - (V_h - V_m)y_{hm}}{k_T}. \tag{12}$$

If a resistance device is installed, the capacitance of the autotransformers and ordinary transformers will change, as follows:

$$\begin{cases} y_{ms0}^{-1} = y_{ms}^{-1} + r \\ y_{ko0}^{-1} = y_{ko}^{-1} + r \end{cases}, \tag{13}$$

where $r$ is the resistance installed at the transformer neutral point. The matrix $\boldsymbol{\Phi}$ is defined to represent the full-network substation effective bias current $\boldsymbol{I}_{eff}$, as follows:

$$\boldsymbol{I}_{eff} = \boldsymbol{\Phi}\boldsymbol{V} = \boldsymbol{\Phi}\boldsymbol{Y}^{-1}\boldsymbol{I}, \tag{14}$$

where $\boldsymbol{\Phi}$ is related to the transformer parameters and whether the neutral point is equipped with a blocking device. In the same substation, the bias current in one transformer will flow to other transformers if only one transformer has an installed CBD, which will aggravate the DC bias of the other transformers. Therefore, when installing DC blocking devices for each substation, all of the main transformers in this substation are assumed to be blocked. Then, $\boldsymbol{I}$, $\boldsymbol{\Phi}$, and $\boldsymbol{Y}$ will change, and the effective bias current of each transformer will change accordingly. Assuming that $S$ CBDs and RCLDs are installed in the system, the effective bias current of the substation is given by the following:

$$\boldsymbol{I}_{eff}(S) = \left(\boldsymbol{\Phi} - \sum_{s \in S}\boldsymbol{\Phi}_s\right)\left(\boldsymbol{Y} - \sum_{s \in S}\boldsymbol{Y}_s\right)^{-1}\boldsymbol{I}, \tag{15}$$

where $\boldsymbol{\Phi}_s$ and $\boldsymbol{Y}_s$ represent the changes in $\boldsymbol{\Phi}$ and $\boldsymbol{Y}$ after the blocking devices are installed in substation $s$.

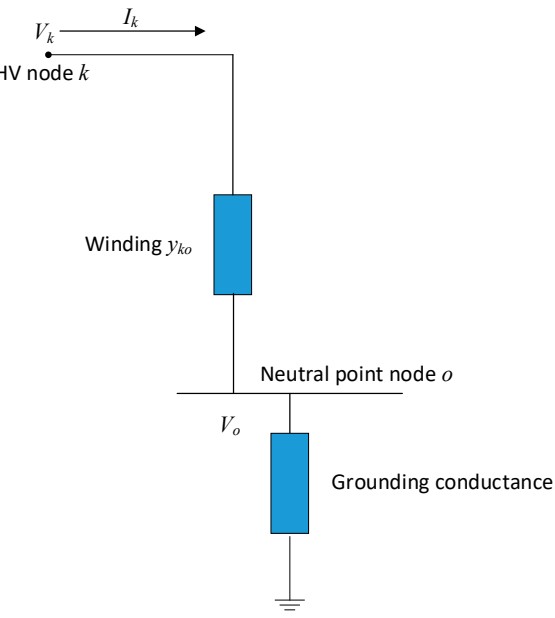

**Figure 2.** Ordinary transformer equivalent circuit model.

*2.3. Combination Optimization Configuration Method for Capacitance and Resistance Devices*

When suppressing the DC bias by installing CBD and RCLD devices at the neutral points of the transformers, the priority is to ensure that the effective bias current of all of the transformers in the

power system does not exceed a set limit. On this basis, the minimum number of devices necessary that must be installed in the entire network can be determined. The objective function is given by the following:

$$\left( \left| I_{eff}^{\max}(S) \right| < A_{\lim} \right) = \min(T_i, i = 1, 2, \ldots S), \tag{16}$$

where $S$ represents the number of installed CBDs or RCLDs, $A_{lim}$ is the set limit, and $T_i$ is the blocking device. The constraint is defined as follows:

$$\left| I_{eff}^{\max}(S) \right| < A_{\lim}. \tag{17}$$

An exhaustive method can be used to solve the above optimal configuration problem when only a few substations exist in the target system. In other words, according to the constraints, all of the configuration schemes can be exhausted to find the global optimal solution. However, the computational cost of the exhaustive method increases rapidly with the increasing number of substations, eventually making the problem impossible to solve. Thus, the problem should be solved using a discrete optimization to rationally select the substation where the device should be installed, considering that numerous substations exist in the equivalent model. In this case, an artificial intelligence method can be adopted, such as genetic [22–24], particle swarm [25,26], or simulated annealing algorithms [27,28]. After researching various model algorithms, we found that a genetic algorithm is particularly suitable for this discrete optimization problem, because it can effectively manage any form of objective function and constraint. A genetic algorithm can also effectively search global information with probabilistic meaning. In this paper, the resistance continuity problem is neglected when selecting the RCLD, and a resistance value of 3 Ω is directly selected and optimized together with the CBD, to illustrate that combination optimization is better than single optimization.

Genetic algorithms evolve better approximate solutions from generation to generation, according to the principle of survival of the fittest after the initial population emerges. In each generation, individuals are selected according to their fitness in the problem domain. The population representing the new solution set is generated by combining crossovers and mutations with genetic operators of natural genetics, and the individual of the new population is closer to the optimal solution. This paper also amends the traditional roulette selection method (Figure 3) to improve the selecting ability of the selection operator. The population size is set as $N$, and the probability of each individual being selected is $P_1, P_2, \ldots, P_N$, where the sum of the selected probabilities is 1. Each round of selections produces a uniform random number between [0, 1] as the selection pointer, and an individual is selected based on where the pointer falls. After $n$ rounds of selection, the probability of individual selection in round $p$ $(0 < p < n)$ is compared, and the individual with the greatest probability is selected. The $p$ value can be adjusted freely according to the actual situation during the selection operator process. This improved roulette selection method not only reduces the errors caused by the randomness, but also maintains the diversity of the population. The overall process is described in Figure 4. First, a random matrix is generated, which is uniformly distributed between [0, 1] and mutually identical to the population matrix. Each element value in the population matrix is determined by each corresponding element value in the random matrix. The elements on each chromosome in the population matrix represent whether the corresponding substation installs a device, as well as what kind of device to install. The initial population size is 80, and the maximum number of iterations is set to 200. Moreover, the crossover probability and mutation probability are 0.9 and 0.2, respectively, where the crossover probability determines whether a crossover operation is performed, and the mutation probability determines whether a mutation occurs in the child. This variation is directly reflected in the transition from the original installation to the other two installations. The new individuals are generated by crossover and mutation, according to the rule that, if the number of installations cannot decrease, the experienced crossover and mutation will be invalid.

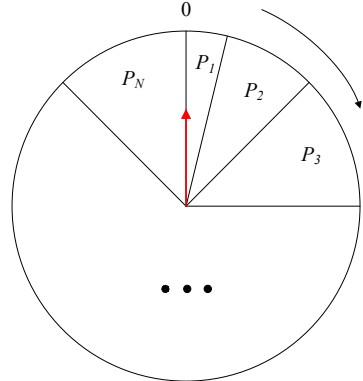

**Figure 3.** Schematic diagram of the roulette selection method.

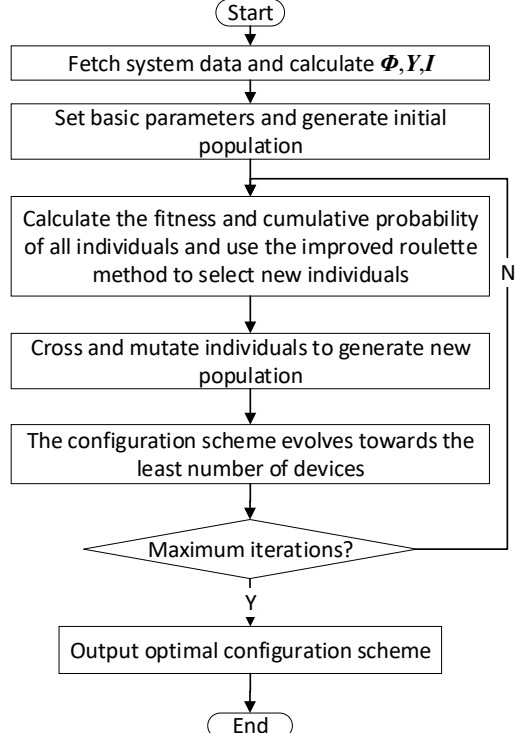

**Figure 4.** Optimal configuration flowchart based on genetic algorithm method.

## 3. Results Analysis

### 3.1. Test Network and Substation Ground Potential Calculation

A DC equivalent model is established using the Xizhe HVDC transmission project receiving-end grid near the Jinsi grounding electrode (Figure 5), which contains 73 substations, 157 nodes, and 187 lines. Each substation is assumed to have two identical autotransformers operating in parallel, both of which have the same maximum effective bias current limit. Additionally, the installation of devices in the substations means that all of the substation transformers are installed in the same condition. The six-layer horizontal uniform soil resistivity parameters of the Jinsi grounding electrode are listed in Table 1, and the main parameters of the receiving-end power grid are shown in Table 2. The grounding current of the DC grounding electrode is set to 5000 A. The finite element method is used to calculate the Earth potential distribution within 150 km of the Jinsi grounding electrode (Figure 5), using ANSYS software.

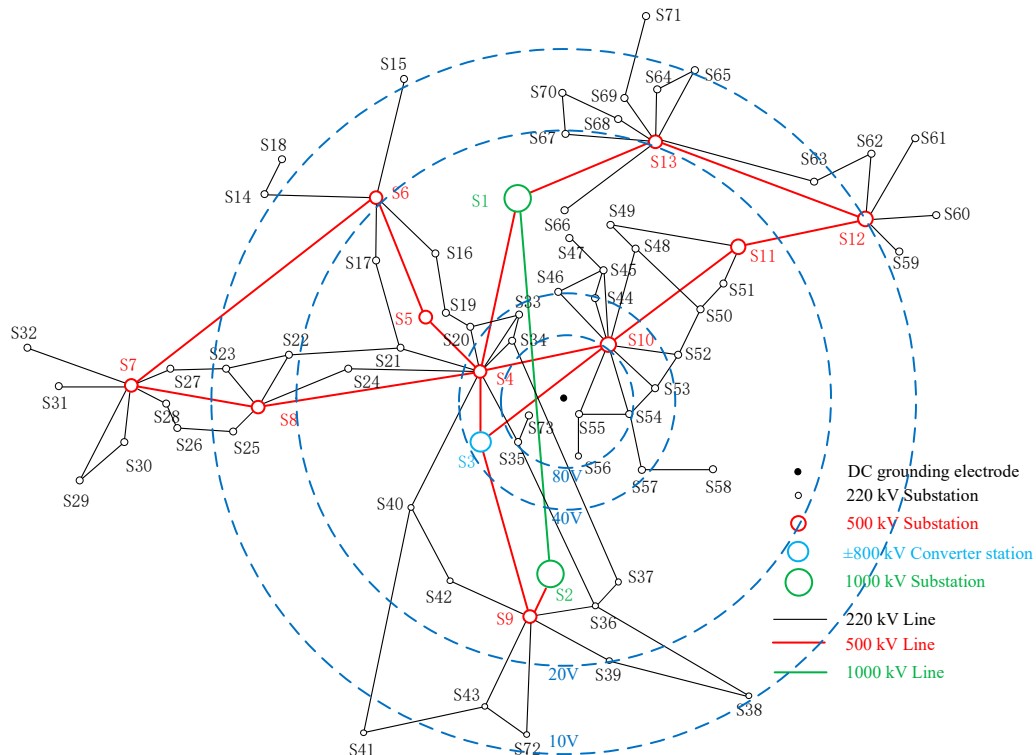

**Figure 5.** Geographical wiring diagram of Xizhe high-voltage direct current (HVDC) transmission project receiving-end grid near the Jinsi grounding electrode.

**Table 1.** Parameters of the layered Earth resistivity model around the Jinsi electrode.

| Layer | Layer Depth/m | Resistivity/Ω·m |
|-------|---------------|-----------------|
| 1 | 0–45 | 50 |
| 2 | 45–155 | 250 |
| 3 | 155–300 | 2500 |
| 4 | 300–600 | 400 |
| 5 | 600–2000 | 1500 |
| 6 | >2000 | 4000 |

**Table 2.** Main direct current (DC) parameters of receiving-end grid. HV—high voltage; MV—medium voltage.

| Voltage Level/kV | Transformer Single-Phase Winding DC Resistance/Ω | | Substation Grounding Resistance/Ω | Line Single-Phase Unit Length DC Resistance/Ω·km$^{-1}$ |
|------------------|------------------|------------------|-----------------|-----------------|
| | **HV Winding** | **MV Winding** | | |
| 220 | 0.45 | - | 0.5 | 0.0748 |
| 500 | 0.238 | 0.097 | 0.2 | 0.0187 |
| 1000 | 0.183 | 0.141 | 0.1 | 0.0095 |

### 3.2. Effect of Devices on Effective Bias Current

Equations (8) and (12) are used to calculate the effective bias current in a single transformer. The effective bias currents of the transformers in the overall network were calculated using matrix Equation (14) for the target grid, without installing CBDs or RCLDs, and the results are shown in Figure 6. The transformers in 500-kV substations 3, 7, 10, and 12, and 220-kV substations 35, 55, 58, and 73 have larger bias currents than the other substation transformers.

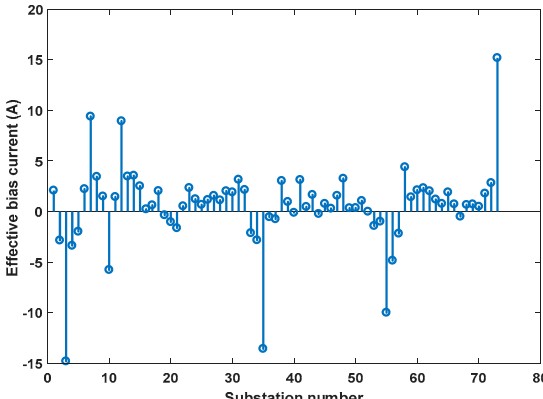

**Figure 6.** Effective bias current in each substation without capacitance blocking devices (CBDs) mounting or resistance current limiting device (RCLDs).

A CBD will only be installed in substations with a large effective bias current if the configuration is not optimized. The effective bias current in an ordinary transformer will be 0 after CBD installation. The maximum effective bias current limit per phase of the transformer is assumed to be 5 A. As shown in Figure 6, substations 3, 7, 10, 12, 35, 55, and 73 require the installation of CBDs, and Figure 7 shows the results of installing CBDs on the effective bias currents of each substation, which were calculated using Equation (15).

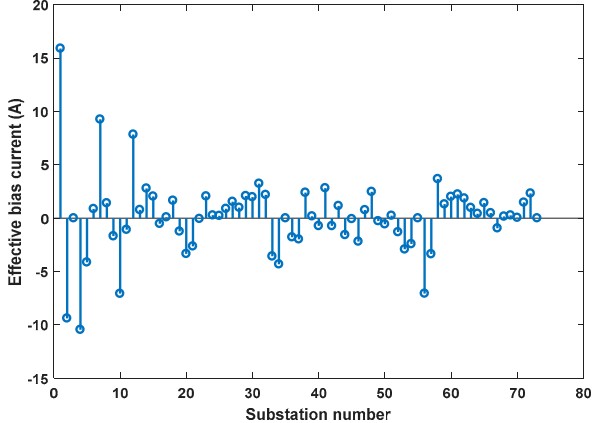

**Figure 7.** Effective bias current in each substation with CBDs.

Figure 7 reveals that the effective bias currents of substations 7, 10, and 12 still exceed the limit of 5 A after installing CBDs, despite the neutral-point current being completely blocked. Thus, the CBDs cannot completely eliminate the transformer DC bias, and thus, using the effective bias current indicator as the control variable is more reasonable. In Figure 7, although the effective bias current of substations 3, 35, 55, and 73 are suppressed after installing CBDs, the effective currents in their adjacent substations increase, especially that of substations 1, 2, 4, and 56, which exceed 5 A after CBD installation. Therefore, only treating the nodes with DC bias currents that exceed the limit can hardly satisfy the system-level requirements, and often increases the bias current in the adjacent transformers. Therefore, the system-wide configuration of such bias current management devices must be discussed. In addition, for economic purposes, we must optimize the number of installed management devices. The optimal configuration will be discussed in the next section.

We further simulate the non-optimal method by installing CBDs in the substations with a large overall effective bias current. Additionally, the maximum transformer effective bias current changes with the number of installed CBDs, as shown in Figure 8.

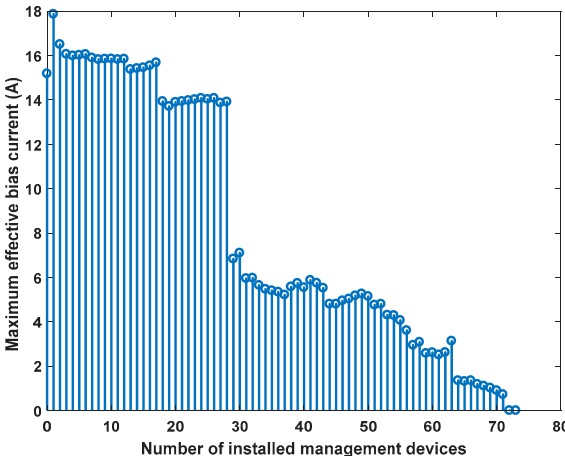

**Figure 8.** Maximum effective bias current as a function of the number of installed CBDs.

Figure 8 reveals that overall, the maximum effective bias current tends to decrease with the increasing number of CBDs. However, the current decreases gently at first, which means installing only a few CBDs at substations with a large bias current does not significantly reduce the maximum effective bias current of the transformers. Thus, the conventional method only has a significant effect when a certain number of CBDs are installed. Moreover, after reaching a critical point, the bias current begins to decrease gradually again, which means that continuing to increase the number of CBDs does not significantly improve the management effect. Thus, if the CBD configuration scheme is not optimized, investment will be wasted, and the operation and maintenance workload will increase in the future.

### 3.3. Optimization Strategy Results Analysis

To determine the minimum number of installed devices, the maximum effective bias current of all substations being less than the limit is taken as the constraint condition. The improved genetic algorithm is applied to configure the number of CBDs installed under different limit currents, and the results are shown in Figure 9, with the non-optimal configuration for comparison. Note that the non-optimal method is a commonly used straightforward solution. Specifically, the non-optimal method decides the priority for installing CBDs by considering the magnitude of the effective bias current. The larger the current is, the higher the priority is.

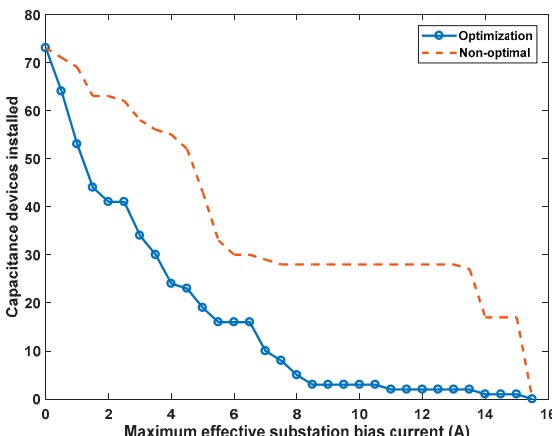

**Figure 9.** Comparison of configurations determined by two methods under different maximum effective bias current limits.

Our optimization method determines that significantly fewer CBDs are required than the non-optimal method. Thus, the optimization configuration method can remarkably reduce the number of CBDs to achieve the same expected level of management, thereby reducing the cost of managing the transformer DC bias. Furthermore, the number of devices determined by the combination optimization, a single RCLD optimization, and a single CBD optimization are compared, in Figure 10, to illustrate the superiority of the combination optimization.

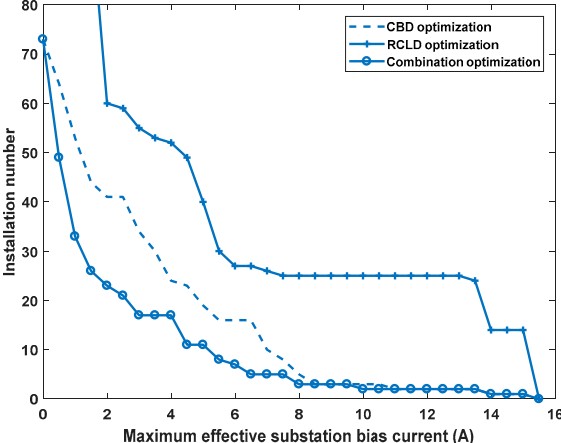

**Figure 10.** Comparison of three optimization configuration methods under different maximum effective bias current limits.

Figure 10 clearly demonstrates that RCLD optimization cannot always restrict the substation maximum effective bias current to a permissible range. Furthermore, installing an excessive number of RCLDs can seriously change the system operation status, thus changing the overall system structure and redistributing the power flow. In that case, the configuration scheme would require further optimization, whereas the CBD optimization does not have such a limitation. However, the single management device optimization methods require more devices to achieve the same management effect compared with the combination optimization, which is economically unfavourable. Our combination optimization method is applicable to any type of AC system, and scalable to other relevant systems, such as geomagnetically induced current (GIC).

## 4. Conclusions

This paper proposed a method for optimizing the configuration of devices for suppressing DC bias in substations. The outcomes were as follows:

The transformer effective bias current is verified to be a more reasonable indicator for describing the influence of the grounding current on transformers, considering that the DC bias current in the autotransformer series winding cannot be ignored when installing a management device at the neutral point. Additionally, this method is applicable to homologous DC bias management issues.

Our optimization method based on the improved genetic algorithm is validated to configure management devices to suppress transformer DC bias in the overall network. Thus, it is essential to optimize the device configuration instead of only treating the nodes with a bias current exceeding the limit. We revealed that the conventional method will increase the bias current of the adjacent transformers.

We demonstrate that the RCLD optimization method does not reduce the effective bias current of the overall system to below the allowable limits. Additionally, the proposed combination optimization method can achieve the ideal management effect with fewer devices than single CBD optimization, which significantly reduces the management costs.

For future work, we could probe the continuity of the resistance value, which can be optimized using the proposed method. In addition, the impact of slight changes in the system structure on the original combination optimization scheme can be analysed.

**Author Contributions:** Conceptualization, C.L.; funding acquisition, C.L.; methodology, D.W.; software, D.W.; writing (original draft), D.W.; writing (review and editing), C.L.

**Funding:** This work was supported by the National Key R&D Program of China (2016YFC0800100).

**Conflicts of Interest:** The authors declare no conflict of interest.

## Nomenclature

HVDC    high-voltage direct current
RCLD    resistance current limiting device
CBD     capacitance blocking device
HV      high voltage
MV      medium voltage
GIC     geomagnetically induced current

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
