# Peer review of "Combination Optimization Configuration Method of Capacitance and Resistance Devices for Suppressing DC Bias in Transformers"

_energies, doi:10.3390/en12091813_

Round 1
Reviewer 1 Report
The authors propose an optimal configuration method for suppressing DC bias in transformers. The simulation results on the Xizhe HVDC transmission project receiving-end grid near the Jinsi grounding electrode are also provided. In general, the paper is well-written and the topic is timely. I think that the quality of this paper can be improved if the authors address the following aspects:
1 The nomenclature should be included to help the reader to follow the paper conveniently.
2 The main contributions and novelty of this paper should be further summarized and clearly demonstrated.
3 How scalable is the proposed method?
4 Add more discussions on Figures 7 and 9 to better verify the work. Even they are informative, more details can enrich the paper.
5 The simulation results should be extended to validate the presented method sufficiently. Authors need to add more results to thoroughly support the main findings.
6 In the conclusion section, this reviewer strongly suggests the authors clearly explain what the significant findings are and why your paper is really important. Some of the most important quantitative results and future directions of the work should also be clearly mentioned in the conclusion part.
7 There are some (very few, indeed) grammatical errors in the paper. Please revise thoroughly.
8 Authors are recommended to include and review the following study to improve the literature survey:
[R1] Two-stage multi-objective OPF for AC/DC grids with VSC-HVDC: Incorporating decisions analysis into optimization process. Energy, 147, pp.286-296, 2018.
[R2] Potential compensation method for restraining the DC bias of transformers during HVDC monopolar operation. IEEE Transactions on Power Delivery, 31(1), 103-111, 2016.
Author Response
Dear Reviewer,
Thank you for your comments concerning our manuscript entitled “Combination Optimization Configuration Method of Capacitance and Resistance Devices for Suppressing DC Bias in Transformers.” These comments have been valuable and helpful for improving our paper. We have carefully studied the comments and have made corresponding corrections in our paper. We hope we have addressed all of your comments.
Regards,
Donghui Wang and Chunming Liu
Comments:
The authors propose an optimal configuration method for suppressing DC bias in transformers. The simulation results on the Xizhe HVDC transmission project receiving-end grid near the Jinsi grounding electrode are also provided. In general, the paper is well-written and the topic is timely. I think that the quality of this paper can be improved if the authors address the following aspects:
1 The nomenclature should be included to help the reader to follow the paper conveniently.
Response: Thank you for your positive comment and valuable suggestion.
We have added a list of all nomenclature before the Introduction section (page 1).
2 The main contributions and novelty of this paper should be further summarized and clearly demonstrated.
Response: Thank you for your valuable comment.
We have summarized the contribution and novelty of this paper in the Introduction section of the revised paper (page 2).
3 How scalable is the proposed method?
Response: Thank you for your valuable question.
The proposed method is applicable to all types of transformers in AC systems. In addition, the optimal strategy can also be applied to other DC bias problems, such as suppressing the geomagnetically induced current in transformers. It can be applied at the system-wide level and is highly scalable. We have discussed this topic in the revised paper.
4 Add more discussions on Figures 7 and 9 to better verify the work. Even they are informative, more details can enrich the paper.
Response: Thank you for your valuable suggestion.
We have added to the discussion of Fig. 7 in section 3.2, line 318, of the revised paper to further enrich the contents of this paper.
We have also added more details to section 3.3, line 374, of the revised paper.
5 The simulation results should be extended to validate the presented method sufficiently. Authors need to add more results to thoroughly support the main findings.
Response: Thank you for your valuable comment.
We have extended the simulation results to sufficiently validate the proposed method in section 3.2, line 331, of the revised paper.
6 In the conclusion section, this reviewer strongly suggests the authors clearly explain what the significant findings are and why your paper is really important. Some of the most important quantitative results and future directions of the work should also be clearly mentioned in the conclusion part.
Response: Thank you for your valuable comment.
We have rewritten the Conclusion section (page 13) of the revised paper to explain the importance of our work and directions for future work. In addition, the most import issues with the proposed method have been discussed.
7 There are some (very few, indeed) grammatical errors in the paper. Please revise thoroughly.
Response: Thank you for your valuable comment.
A native English speaker and professional editor has reviewed the entire manuscript, including the revisions discussed herein. The grammar and punctuation have been thoroughly revised.
8 Authors are recommended to include and review the following study to improve the literature survey:
[R1] Two-stage multi-objective OPF for AC/DC grids with VSC-HVDC: Incorporating decisions analysis into optimization process. Energy, 147, pp.286-296, 2018.
[R2] Potential compensation method for restraining the DC bias of transformers during HVDC monopolar operation. IEEE Transactions on Power Delivery, 31(1), 103-111, 2016.
Response: Thank you for your valuable comment and suggestions.
We have added references [18] and [19] to the Reference list of the revised paper (page 14) to improve the literature survey. We have also added the following sentence to the Introduction section:
A potential compensation method for restraining the DC bias of autotransformers is discussed in [18, 19].
[18] Li Y.; Li Y.H.; Li G.Q.; Zhao D.B.; Chen C. Two-stage multi-objective OPF for AC/DC grids with VSC-HVDC: Incorporating decisions analysis into optimization process. Energy, 2018, 147, 286-296. DOI: 10.1016/j.energy.2018.01.036.
[19] Pan Z.H.; Wang X.M.; Tan B.; Zhu L.; Liu Y.; Liu Y.L.; Wen X.S. Potential compensation method for restraining the DC bias of transformers during HVDC monopolar operation. IEEE Transactions on Power Delivery, 2016, 31, 103-111. DOI: 10.1109/TPWRD.2015.2438853

Reviewer 2 Report
The paper demonstrated an optimization process of a CBD and a RCLD combined DC bias suppressing technique for a transformer network.
The paper needs a huge improvement in its presentation details to be accepted.
The paper needs to check English expressions and grammars.
The title of section 2 was "Analyzing the impact of ground current on transformers", however, there was no analysis result showing the impact of the ground current on transformer.
The title of section 2.1 was "Earth potential distribution around DC grounding electrode", however, there was no result showing the earth potential distribution around an electrode. This discrepancy between the title and the text body was making the reader very difficult to follow the idea of the paper.
Furthermore, the authors showed some equations without references or any supporting comments. For example, the equations (4) , (5), and (6) showed up without any reference and the authors did not explain how did they get the equations by.
Also, Ie was not shown in Figure 1, therefore it was not able to check if equation (4) was correct or not.
Also, the reference point of Vh, Vm, Vs are not shown in Figure 1. Therefore, it was not clear whether the voltages are the potential relative to the ground or to the neutral point. It was not helpful in understanding the paper.
There were typos, i.e. on line 104, " medium voltage node j" -> there was no node j in the figure.
In addition, there was no explain on the difference between figure 1 and figure 2 and Ie and IE. Why figure 2 (ordinary transformer) pops up suddenly without any explanation ?
Equations (8), (9), and (10) also presented without any background information or reference. No reader can understand them in this paper.
Please show the process how the authors get the effective bias current in Figure 6. Did you use the equations in (1) to (10)? Please make it clear.
Please show the process how the authors get the new effective bias current in Figure 7.
For figure 8, how the authors get the non-optimal configurations? What was the non-optimal solutions?
My final comment is about the evaluation of the proposed idea. Is there any way that can prove your idea? Without any proof, the paper cannot be accepted.
Author Response
Dear Reviewer,
Thank you for your comments concerning our manuscript entitled “Combination Optimization Configuration Method of Capacitance and Resistance Devices for Suppressing DC Bias in Transformers.” These comments have been valuable and helpful for improving our paper. We have carefully studied the comments and have made corresponding corrections in our paper. We hope we have addressed all of your comments.
Regards,
Donghui Wang and Chunming Liu
Comments:
The paper demonstrated an optimization process of a CBD and a RCLD combined DC bias suppressing technique for a transformer network.
The paper needs a huge improvement in its presentation details to be accepted.
The paper needs to check English expressions and grammars.
Response: Thank you for your valuable comments.
A native English speaker and professional editor has reviewed the entire manuscript, including the revisions discussed herein. The grammar and punctuation have been thoroughly revised.
The title of section 2 was "Analyzing the impact of ground current on transformers", however, there was no analysis result showing the impact of the ground current on transformer.
Response: Thank you for your valuable comment.
We have renamed section 2 “Methodology”.
The title of section 2.1 was "Earth potential distribution around DC grounding electrode", however, there was no result showing the earth potential distribution around an electrode. This discrepancy between the title and the text body was making the reader very difficult to follow the idea of the paper.
Response: Thank you for your valuable comment.
We have renamed section 2.1 “Earth Potential Distribution Calculation Method”.
Furthermore, the authors showed some equations without references or any supporting comments. For example, the equations (4), (5), and (6) showed up without any reference and the authors did not explain how did they get the equations by.
Response: Thank you for your valuable comment.
Equations (4), (5), and (6) are obtained from the autotransformer equivalent circuit.
Equation (4) means the transformer effective bias current Ie is the sum of the DC current both at the high-voltage side Ih and the medium-voltage side Im, which should be multiplied by the ratio of turns to convert to the high-voltage side.
According to the theory of the node admittance matrix, the current can be converted to the product of voltage and conductance. Va is the node voltage matrix, and the matrix Φa is the corresponding conductivity of each winding after being converted to the high-voltage side. The derivation process is based on the circuit shown in Fig. 1. We have thoroughly checked the formulas to ensure the formulation is complete.
Also, Ie was not shown in Figure 1, therefore it was not able to check if equation (4) was correct or not.
Response: Thank you for your valuable comment.
Ie is the effective bias current, which is an abstract variable we defined. It is the sum of the DC current at both the high-voltage side Ih and the medium-voltage side Im converted to the high-voltage side. We define Ie because it is a better indicator for describing the influence of the grounding current on transformers.
Also, the reference point of Vh, Vm, Vs are not shown in Figure 1. Therefore, it was not clear whether the voltages are the potential relative to the ground or to the neutral point. It was not helpful in understanding the paper.
Response: Thank you for your valuable comment. We are very sorry for the insufficient description.
The reference point is Vs, which we have clarified in the revised paper.
There were typos, i.e. on line 104, " medium voltage node j" -> there was no node j in the figure.
Response: Thank you for your valuable comment. We are very sorry for the error, which we corrected. The medium-voltage node is node m.
In addition, there was no explain on the difference between figure 1 and figure 2 and Ie and IE. Why figure 2 (ordinary transformer) pops up suddenly without any explanation?
Response: Thank you for your valuable comment. We are very sorry for the insufficient explanation.
For an ordinary transformer that is not an autotransformer, there is no electrical link between the high- and low-voltage side; thus, the effective bias current flows through the neutral point. We have explained this in the revised paper.
Equations (8), (9), and (10) also presented without any background information or reference. No reader can understand them in this paper.
Response: Thank you for your valuable comment. We are very sorry for the insufficient information.
We have rewritten Equations (8), (9), and (10) to avoid ambiguity.
When a DC current is injected into the ground through an electrode, a potential distribution forms around the electrode. The bias currents are driven toward transformers (through transmission lines) when their neutral-point potentials are different. Only the resistances need to be considered for a DC circuit. When the CBD is not installed at the neutral point d of a transformer, the current Id injected into the neutral point d is:
, (8)
where Vd represents the node d ground potential caused by the grounding electrode. Rd is the ground resistance corresponding to node d. The relationship between the node voltage vector V and the node injection current vector I is:
, (9)
where Y represents the network node admittance matrix. We also deleted Equation (10).
Please show the process how the authors get the effective bias current in Figure 6. Did you use the equations in (1) to (10)? Please make it clear.
Response: Thank you for your valuable comment.
Fig. 6 shows the original effective bias currents in each substation without installing any devices.
Equations (11) and (7) are used to calculate the effective bias current in a single transformer. Considering there are many transformers in the target network, we use matrix Equation (13) to calculate the effective bias currents in the overall network. We have clarified this point.
Please show the process how the authors get the new effective bias current in Figure 7.
Response: Thank you for your valuable suggestion.
Fig. 7 shows the effective bias currents in each substation with CBDs installed. The effective bias current in an ordinary transformer will be 0 after CBD installation. We use matrix Equation (14) to calculate the transformer effective bias currents in the overall network after installing CBDs. Note that only substations 3, 7, 10, 12, 35, 55, and 73 have CBDs installed in Fig. 7. We have added the necessary explanation to clarify this point.
For figure 8, how the authors get the non-optimal configurations? What was the non-optimal solutions?
Response: Thank you for your valuable comment.
The non-optimal method is a commonly used straightforward solution that decides the priority for installing CBDs based on the magnitude of the effective bias current. The larger the current is, the higher the priority is. We have explained this point in the revised paper.
My final comment is about the evaluation of the proposed idea. Is there any way that can prove your idea? Without any proof, the paper cannot be accepted.
Response: In the “Results analysis” section, we use the case of a practical project to prove our that our optimization method is better than the traditional non-optimal method. We compared the results obtained from the two methods, and Fig. 9 and Fig. 10 reveal that our proposed method requires fewer devices to suppress the bias current to the target level. We have emphasized this difference in the revised paper.

Reviewer 3 Report
This paper deals with proposing an optimal configuration method for suppressing DC bias in the substation. The novelty of this work and the authors’ contribution to the existing method are not clear. Also, there are some critical issues in this paper which must be addressed/modified, precisely.
1- The paper needs major proofreading. In addition, punctuation should be checked, again. All abbreviations have to be explained once and in order. The keywords should be in alphabetical order.
2- The literature review insufficient and inconsistent. The papers which are studied should be sorted and the problem(s) of each section should be defined. The authors should clearly define their problem, as well. A comprehensive literature review based on recent reference papers is required. Moreover, the authors should use a narrative style based on recent research works. There are many papers that the authors can read them and improve the quality of their work.
3- The configuration of the HVDC system is not well-studied and explained in this paper. Kindly refer the following paper for the HVDC transmission system configuration [https://www.researchgate.net/publication/331396618_Power_Management_Strategy_in_Multi-Terminal_VSC-HVDC_System].
4- The problem formulation is not complete. Also, there are some missing parameters in the problem formulation. Parameters are used but not defined. Therefore, the authors are advised to check the formulas line-by-line and define the used parameters.
5- The authors have not provided a comparison with other recent research works. Still, the reviewer does not see any superiority of this work over the existing methods.
Author Response
Dear Reviewer,
Thank you for your comments concerning our manuscript entitled “Combination Optimization Configuration Method of Capacitance and Resistance Devices for Suppressing DC Bias in Transformers.” These comments have been valuable and helpful for improving our paper. We have carefully studied the comments and have made corresponding corrections in our paper. We hope we have addressed all of your comments.
Regards,
Donghui Wang and Chunming Liu
Comments:
This paper deals with proposing an optimal configuration method for suppressing DC bias in the substation. The novelty of this work and the authors’ contribution to the existing method are not clear. Also, there are some critical issues in this paper which must be addressed/modified, precisely.
1- The paper needs major proofreading. In addition, punctuation should be checked, again. All abbreviations have to be explained once and in order. The keywords should be in alphabetical order.
Response: Thank you for your valuable comment.
A native English speaker and professional editor has reviewed the entire manuscript, including the revisions discussed herein. The grammar and punctuation have been thoroughly revised. We also added a list of all nomenclature before the Introduction section (page 1), and the keywords are now listed in alphabetical order (page 1).
2- The literature review insufficient and inconsistent. The papers which are studied should be sorted and the problem(s) of each section should be defined. The authors should clearly define their problem, as well. A comprehensive literature review based on recent reference papers is required. Moreover, the authors should use a narrative style based on recent research works. There are many papers that the authors can read them and improve the quality of their work.
Response: Thank you for your valuable comment.
We have thoroughly studied recent papers, and we have sorted these papers and defined the corresponding problems. In addition, we have reviewed the style of the paper for consistency with related reports. Lastly, we have rewritten the literature review in the Introduction section (page 2).
3- The configuration of the HVDC system is not well-studied and explained in this paper. Kindly refer the following paper for the HVDC transmission system configuration [https://www.researchgate.net/publication/331396618_Power_Management_Strategy_in_Multi-Terminal_VSC-HVDC_System].
Response: Thank you for your valuable suggestion.
We have included reference [4] in the Reference list of the revised paper (page 14) to comprehensively explain the configuration of the HVDC system. We have also added the following sentence in the Introduction section (page 2):
The configuration of such a HVDC transmission system was studied in [4].
[4] Mohammadi F. Power Management Strategy in Multi-Terminal VSC-HVDC System. 4th National Conference on Applied Research in Electrical, Mechanical Computer and IT Engineering. 2018. [https://www.researchgate.net/publication/331396618_Power_Management_Strategy_in_Multi-Terminal_VSC-HVDC_System].
4- The problem formulation is not complete. Also, there are some missing parameters in the problem formulation. Parameters are used but not defined. Therefore, the authors are advised to check the formulas line-by-line and define the used parameters.
Response: Thank you for your valuable comment.
We have thoroughly checked the formulas line by line. We have added the relevant parameters to adequately explain the problem in section 2.2 of the revised paper.
5- The authors have not provided a comparison with other recent research works. Still, the reviewer does not see any superiority of this work over the existing methods.
Response: Thank you for your valuable comment.
We have compared our results with recent works, and we have highlighted the contribution and novelty of this paper in the Introduction section of the revised paper (page 2).
Additionally, we have rewritten the Conclusion section (page 13) of the revised paper to explain the importance of this work and the superiority of our method.

Round 2
Reviewer 1 Report
Most of my comments have addressed properly. The revision is appropriate. I suggest accepting for publication.
Author Response
Dear Reviewer, Thank you for your comments concerning our manuscript entitled “Combination Optimization Configuration Method of Capacitance and Resistance Devices for Suppressing DC Bias in Transformers.” These comments have been valuable and helpful for improving our paper. We have carefully studied the comments and have made corresponding corrections in our paper. We hope we have addressed all of your comments. Regards, Donghui Wang and Chunming Liu Comments: Most of my comments have addressed properly. The revision is appropriate. I suggest accepting for publication. Response: Thank you for your positive comments.

Reviewer 2 Report
The paper demonstrated an optimization process of a CBD and a RCLD combined DC bias suppressing technique for a transformer network.
The paper needs a huge improvement in its presentation details to be accepted.
The paper needs to check English expressions and grammars.
Response: Thank you for your valuable comments.
A native English speaker and professional editor has reviewed the entire manuscript, including the revisions discussed herein. The grammar and punctuation have been thoroughly revised.
The title of section 2 was "Analyzing the impact of ground current on transformers", however, there was no analysis result showing the impact of the ground current on transformer.
Response: Thank you for your valuable comment.
We have renamed section 2 “Methodology”.
My point was not the title. The authors used equation (1), (2) and (3) to calculate the earth potential, but there was no evaluation of the equations whether the calculated potentials were correct or not. That is, the equations were shown in the paper without any analysis and evaluation on them. These was the same with the rest of the equation used in the paper.
The title of section 2.1 was "Earth potential distribution around DC grounding electrode", however, there was no result showing the earth potential distribution around an electrode. This discrepancy between the title and the text body was making the reader very difficult to follow the idea of the paper.
Response: Thank you for your valuable comment.
We have renamed section 2.1 “Earth Potential Distribution Calculation Method”.
Furthermore, the authors showed some equations without references or any supporting comments. For example, the equations (4), (5), and (6) showed up without any reference and the authors did not explain how did they get the equations by.
Response: Thank you for your valuable comment.
Equations (4), (5), and (6) are obtained from the autotransformer equivalent circuit.
Equation (4) means the transformer effective bias current Ie is the sum of the DC current both at the high-voltage side Ih and the medium-voltage side Im, which should be multiplied by the ratio of turns to convert to the high-voltage side.
According to the theory of the node admittance matrix, the current can be converted to the product of voltage and conductance. Va is the node voltage matrix, and the matrix Φa is the corresponding conductivity of each winding after being converted to the high-voltage side. The derivation process is based on the circuit shown in Fig. 1. We have thoroughly checked the formulas to ensure the formulation is complete.
Again, the equations (4)-(6) used without any evaluation on them. How can you sure that the equations are correct for modeling in the real system?
And, I am not sure why the authors used the admittance for the DC currents. If Ih and Im are DC currents, is there any reason for using the admittance not just the resistances?
Also, Ie was not shown in Figure 1, therefore it was not able to check if equation (4) was correct or not.
Response: Thank you for your valuable comment.
Ie is the effective bias current, which is an abstract variable we defined. It is the sum of the DC current at both the high-voltage side Ih and the medium-voltage side Im converted to the high-voltage side. We define Ie because it is a better indicator for describing the influence of the grounding current on transformers.
Please add Ie on Figure 1.
Also, the reference point of Vh, Vm, Vs are not shown in Figure 1. Therefore, it was not clear whether the voltages are the potential relative to the ground or to the neutral point. It was not helpful in understanding the paper.
Response: Thank you for your valuable comment. We are very sorry for the insufficient description.
The reference point is Vs, which we have clarified in the revised paper.
Please provide the number of lines and pages. It is still unclear that the Vh is defined relative to Vs in Figure 1.
There were typos, i.e. on line 104, " medium voltage node j" -> there was no node j in the figure.
Response: Thank you for your valuable comment. We are very sorry for the error, which we corrected. The medium-voltage node is node m.
In addition, there was no explain on the difference between figure 1 and figure 2 and Ie and IE. Why figure 2 (ordinary transformer) pops up suddenly without any explanation?
Response: Thank you for your valuable comment. We are very sorry for the insufficient explanation.
For an ordinary transformer that is not an autotransformer, there is no electrical link between the high- and low-voltage side; thus, the effective bias current flows through the neutral point. We have explained this in the revised paper.
Please provide the number of lines and pages that you revised.
Equations (8), (9), and (10) also presented without any background information or reference. No reader can understand them in this paper.
Response: Thank you for your valuable comment. We are very sorry for the insufficient information.
We have rewritten Equations (8), (9), and (10) to avoid ambiguity.
When a DC current is injected into the ground through an electrode, a potential distribution forms around the electrode. The bias currents are driven toward transformers (through transmission lines) when their neutral-point potentials are different. Only the resistances need to be considered for a DC circuit. When the CBD is not installed at the neutral point d of a transformer, the current Id injected into the neutral point d is:
, (8)
where Vd represents the node d ground potential caused by the grounding electrode. Rd is the ground resistance corresponding to node d. The relationship between the node voltage vector V and the node injection current vector I is:
, (9)
where Y represents the network node admittance matrix. We also deleted Equation (10).
Please show the process how the authors get the effective bias current in Figure 6. Did you use the equations in (1) to (10)? Please make it clear.
Response: Thank you for your valuable comment.
Fig. 6 shows the original effective bias currents in each substation without installing any devices.
Equations (11) and (7) are used to calculate the effective bias current in a single transformer. Considering there are many transformers in the target network, we use matrix Equation (13) to calculate the effective bias currents in the overall network. We have clarified this point.
Please add this comment in the paper.
Please show the process how the authors get the new effective bias current in Figure 7.
Response: Thank you for your valuable suggestion.
Fig. 7 shows the effective bias currents in each substation with CBDs installed. The effective bias current in an ordinary transformer will be 0 after CBD installation. We use matrix Equation (14) to calculate the transformer effective bias currents in the overall network after installing CBDs. Note that only substations 3, 7, 10, 12, 35, 55, and 73 have CBDs installed in Fig. 7. We have added the necessary explanation to clarify this point.
Please provide the number of lines and pages that you modified.
For figure 8, how the authors get the non-optimal configurations? What was the non-optimal solutions?
Response: Thank you for your valuable comment.
The non-optimal method is a commonly used straightforward solution that decides the priority for installing CBDs based on the magnitude of theeffective bias current. The larger the current is, the higher the priority is. We have explained this point in the revised paper.
My final comment is about the evaluation of the proposed idea. Is there any way that can prove your idea? Without any proof, the paper cannot be accepted.
Response: In the “Results analysis” section, we use the case of a practical project to prove our that our optimization method is better than the traditional non-optimal method. We compared the results obtained from the two methods, and Fig. 9 and Fig. 10 reveal that our proposed method requires fewer devices to suppress the bias current to the target level. We have emphasized this difference in the revised paper.
This is the most important point. Please provide any experimental result that can show your optimization result is valuable. Not just the simulation or calculation results.
Author Response
Dear Reviewer,
Thank you for your comments concerning our manuscript entitled “Combination Optimization Configuration Method of Capacitance and Resistance Devices for Suppressing DC Bias in Transformers.” These comments have been valuable and helpful for improving our paper. We have carefully studied the comments and have made corresponding corrections in our paper. We hope we have addressed all of your comments.
Regards,
Donghui Wang and Chunming Liu
Comments:
My point was not the title. The authors used equation (1), (2) and (3) to calculate the earth potential, but there was no evaluation of the equations whether the calculated potentials were correct or not. That is, the equations were shown in the paper without any analysis and evaluation on them. These was the same with the rest of the equation used in the paper.
Response: Thank you for your valuable comment.
Our methods presented in equations (1), (2) and (3) are based on inferences of Maxwell's Equations. The calculation method of earth potential near HVDC grounding electrode has been thoroughly discussed in previous works (e.g. [2, 12]). We have clarified this point in section 2.1, line 127, of the revised paper.
Again, the equations (4)-(6) used without any evaluation on them. How can you sure that the equations are correct for modeling in the real system?
Response: Thank you for your valuable comment.
We have rewritten the equations (4)-(6). The autotransformer equivalent circuit is generally modelled as in Fig. 1. The effective bias current Ie on autotransformer is the variable we defined in Equation (4) considering its magnetic flux. The transformer bias current calculation is based on the autotransformer equivalent circuit (Fig. 1). Equations (5), (6), and (7) are obtained from the equivalent circuit to calculate the effective bias current. We have explained this in section 2.2, line 150, of the revised paper.
And, I am not sure why the authors used the admittance for the DC currents. If Ih and Im are DC currents, is there any reason for using the admittance not just the resistances?
Response: Thank you for your valuable comment.
The resistances are used for the DC currents. We prefer the admittance because it is more general to use the admittance in circuit calculation.
Please add Ie on Figure 1
Response: Thank you for your valuable comment.
The effective bias current Ie on Figure 1 is the variable we defined. It is the sum of the DC current at both the HV side Ih and at the MV side Im, which should be multiplied by the ratio of turns to convert to the HV side. It cannot be represented by the current on one component.
Please provide the number of lines and pages. It is still unclear that the Vh is defined relative to Vs in Figure 1.
Response: Thank you for your valuable comment.
The neutral-point node is s which is the reference point. Its potential is Vs. We have clarified this in section 2.2, line 163, of the revised paper.
Please provide the number of lines and pages that you revised.
Response: Thank you for your valuable comment.
We have explained this in section 2.2, line 177, of the revised paper.
Please add this comment in the paper.
Response: Thank you for your valuable comment.
We have added this comment in section 3.2, line 304, of the revised paper.
Please provide the number of lines and pages that you modified
Response: Thank you for your valuable comment.
We have modified this in section 3.2, line 314, of the revised paper.
This is the most important point. Please provide any experimental result that can show your optimization result is valuable. Not just the simulation or calculation results.
Response: Thank you for your valuable comment.
The parameters and geographical diagram of the Xizhe HVDC transmission project receiving-end grid were obtained from practical projects. This optimization method is to address the practical issue. The conventional method was previously adopted to suppress transformer DC bias in the 500 kV substations, which subsequently increased the DC bias in the 220 kV substations. Our optimization method was successfully applied to suppress transformer DC bias in the target network near the Jinsi grounding electrode without increasing the bias current of adjacent transformers. We have validated the optimization method in section 3.

Reviewer 3 Report
My concerns are addressed, and I got no more comments. The only issue is that for all the references which no DOI is available, kindly DO NOT use the link of papers. For example, ResearchGate updates its server every three months and the readers may not have access to the paper that the authors have cited. Using the name of the author(s), the title of the paper, name of journal/conference, and date is enough. This is the standard version of citing papers.
Author Response
Dear Reviewer,
Thank you for your comments concerning our manuscript entitled “Combination Optimization Configuration Method of Capacitance and Resistance Devices for Suppressing DC Bias in Transformers.” These comments have been valuable and helpful for improving our paper. We have carefully studied the comments and have made corresponding corrections in our paper. We hope we have addressed all of your comments.
Regards,
Donghui Wang and Chunming Liu
Comments:
My concerns are addressed, and I got no more comments. The only issue is that for all the references which no DOI is available, kindly DO NOT use the link of papers. For example, ResearchGate updates its server every three months and the readers may not have access to the paper that the authors have cited. Using the name of the author(s), the title of the paper, name of journal/conference, and date is enough. This is the standard version of citing papers.
Response: Thank you for your positive comments and valuable suggestions.
We have revised reference [4] in the Reference list of the revised paper (page 14).
[4] Mohammadi F. Power Management Strategy in Multi-Terminal VSC-HVDC System. 4th National Conference on Applied Research in Electrical, Mechanical Computer and IT Engineering. 2018.
